# Microbial eukaryotes have adapted to hypoxia by horizontal acquisitions of a gene involved in rhodoquinone biosynthesis

Courtney W Stairs[1†], Laura Eme[1†], Sergio A Muñoz-Gómez[1], Alejandro Cohen[2], Graham Dellaire[3,4], Jennifer N Shepherd[5], James P Fawcett[2,6,7], Andrew J Roger[1]*

[1]Centre for Comparative Genomics and Evolutionary Bioinformatics (CGEB), Department of Biochemistry and Molecular Biology, Dalhousie University, Halifax, Canada; [2]Proteomics Core Facility, Life Sciences Research Institute, Dalhousie University, Halifax, Canada; [3]Department of Pathology, Dalhousie University, Halifax, Canada; [4]Department of Biochemistry and Molecular Biology, Dalhousie University, Halifax, Canada; [5]Department of Chemistry and Biochemistry, Gonzaga University, Spokane, United States; [6]Department of Pharmacology, Dalhousie University, Halifax, Canada; [7]Department of Surgery, Dalhousie University, Halifax, Canada

*For correspondence:
Andrew.Roger@Dal.Ca

Present address: [†]Department of Cell and Molecular Biology, Uppasala University, Uppsala, Sweden

Competing interests: The authors declare that no competing interests exist.

**Abstract** Under hypoxic conditions, some organisms use an electron transport chain consisting of only complex I and II (CII) to generate the proton gradient essential for ATP production. In these cases, CII functions as a fumarate reductase that accepts electrons from a low electron potential quinol, rhodoquinol (RQ). To clarify the origins of RQ-mediated fumarate reduction in eukaryotes, we investigated the origin and function of *rquA*, a gene encoding an RQ biosynthetic enzyme. *RquA* is very patchily distributed across eukaryotes and bacteria adapted to hypoxia. Phylogenetic analyses suggest lateral gene transfer (LGT) of *rquA* from bacteria to eukaryotes occurred at least twice and the gene was transferred multiple times amongst protists. We demonstrate that RquA functions in the mitochondrion-related organelles of the anaerobic protist *Pygsuia* and is correlated with the presence of RQ. These analyses reveal the role of gene transfer in the evolutionary remodeling of mitochondria in adaptation to hypoxia.
DOI: https://doi.org/10.7554/eLife.34292.001

## Introduction

In aerobic eukaryotes, complexes I (CI) and II (CII) of the mitochondrial electron transport chain (ETC) oxidize reduced cofactors generated by the tricarboxylic acid (TCA) cycle (NADH and FADH$_2$ respectively) and reduce the lipid-soluble electron carrier ubiquinone (UQ) to ubiquinol (UQH$_2$). Complex III (CIII) oxidizes UQH$_2$ and transfers electrons to cytochrome c oxidase (Complex IV, CIV; via cytochrome c) and eventually to O$_2$, converting it to H$_2$O (*Figure 1A*). In the process, CI, CIII, and CIV generate a proton gradient across the inner mitochondrial membrane that fuels oxidative phosphorylation of ADP to ATP by the F$_1$-F$_o$ ATP synthase (Complex V, CV). However, the ETCs of facultative anaerobes have adapted to function in the absence of oxygen by utilizing alternative terminal electron acceptors (*Kita et al., 1988*; *Tielens, 1994*; *Müller et al., 2012*). For example, in the adult stage of *Ascaris suum*, CIII and CIV are down-regulated but CI continues to oxidize NADH and pump protons (*Sakai et al., 2012*). In this organism, CI transfers electrons from NADH to the

**Figure 1.** Structure and function of ubiquinone and rhodoquinone in mitochondria. (A) Under aerobic conditions, electrons from NADH or succinate (Suc) are shuttled to ubiquinone (UQ) via Complex I (I) or Complex II (II) respectively to generate NAD (NAD+), fumarate (fum) and ubiquinol (UQH2). The electron transfer via Complex I fuels the transport of protons from the mitochondrial matrix into the intermembrane space (IMS). Electrons from UQH$_2$ are transferred to Complex III, cytochrome c, and ultimately O$_2$ via Complex IV with the concomitant pumping of protons. Complex V (V) uses this proton gradient to synthesize ATP. (B) In anaerobic eukaryotes, electrons from NADH are shuttled to rhodoquinone (RQ) to generate rhodoquinol (RH2). The RQ pool is regenerated via CII functioning as a fumarate reductase.

DOI: https://doi.org/10.7554/eLife.34292.002

quinone, rhodoquinone (RQ), generating rhodoquinol (RQH$_2$). RQH$_2$ is reoxidized by CII functioning as a fumarate reductase (FRD), reducing fumarate to succinate. Thus, the adult *Ascaris* mitochondrial ETC is still able to generate a proton gradient to fuel ATP synthesis, even in the absence of oxygen (*Figure 1B*). RQ is structurally similar to UQ but possesses an amino group instead of a methoxy group on the quinone ring (*Figure 1B*). The lower electron potential of RQ ($-63$ mV), compared to UQ ($+100$ mV), favors the FRD reaction (*Castro-Guerrero et al., 2005*). RQ has also been detected in other eukaryotes that experience hypoxia including *Caenorhabditis elegans* (*Takamiya et al., 1999*), parasitic helminths (*Kita et al., 1988*; *Van Hellemond et al., 1996*), *Euglena gracilis* (*Castro-Guerrero et al., 2005*) and *Nyctotherus ovalis* (*Boxma et al., 2005*). However, the presence of RQ in mitochondria or related organelles of other anaerobic eukaryotes has not been established.

Mitochondrion-related organelles (MROs) are specialized mitochondria found in anaerobic protistan lineages that have evolved to function in low oxygen conditions (*Stairs et al., 2015*). The properties of MROs vary between organisms ranging from the simple 'mitosomes' of *Giardia intestinalis*, that appear to function solely in iron-sulfur (Fe-S) cluster generation, to the 'hydrogenomes' of *Trichomonas vaginalis*, or 'hydrogen-producing mitochondria' of *Nyctotherus ovalis* that couple ATP generation to hydrogen production (for review see *Stairs et al., 2015*; *Müller et al., 2012*). Of the various adaptations to anaerobiosis, the reduction or remodeling of the respiratory chain is of central importance to the ATP-generating function of the resulting organelles. Some anaerobic protists have completely lost all components of the respiratory chain, while others have maintained all or parts it (*Stairs et al., 2015;Gawryluk et al., 2016*). While RQ has been proposed to function as an electron carrier in the respiratory chains of some of these protists (*Müller et al., 2012*; *Gawryluk et al., 2016*), the presence of RQ has not been described. Recently, biochemical and genetic investigations of the alphaproteobacterium *Rhodospirillum rubrum* have demonstrated that UQ is a precursor to RQ (*Brajcich et al., 2010*), and that a putative methyltransferase (since named RquA) is essential for RQ biosynthesis and anaerobic growth (*Lonjers et al., 2012*). Here, we explore the distribution, evolutionary origins of RquA and RQ in microbial eukaryotes.

## Results

### The phylogenetic distribution of rquA

RquA homologs were retrieved from various publicly available databases using the *Rhodospirillum rubrum* sequence as the query in sequence similarity searches. We retrieved a total of 182 sequences from unique taxonomic units (i.e., NCBI taxonomy ID) and reduced this dataset to 166 sequences based on a sequence identity cutoff of less than 90%. The *rquA* gene is extremely rare in bacteria and eukaryotes, and we could not identify homologs in any publicly available archaeal genomes. *RquA* genes are found among sparse representatives of only seven different orders of alpha-, beta- and gammaproteobacteria (Burkholderiales, Magnetococcales, Neisseriales, Rhizobiales, Rhodobacteriales, Rhodocyclales and Rhodospirillales). To investigate the phyletic distribution of *rquA* in alphaproteobacteria specifically, we performed a phylogenomic analysis of *rquA*-containing genomes and their *rquA*-lacking relatives (discussed below).

Within eukaryotes, we identified *rquA* homologs in 24 representatives of four of the five supergroups of eukaryotes (Obazoa, Amoebozoa, Sar, and Excavata; *Supplementary file 1*). The gene could not be found in the vast majority of eukaryotic genomes and transcriptome surveys available on Genbank (summarized in *Supplementary file 1*). Note that the absence of *rquA* in some of these data (particularly the transcriptomes) may be due to a lack of depth of sequence sampling. Within breviates, we identified *rquA* in *Pygsuia biforma,* but not in its close relatives, *Breviata anathema* and *Lenisia limosa*, despite the ample genomic and transcriptomic data available for the latter (*Hamann et al., 2016*). We identified spliceosomal introns in the *rquA* genes in eukaryotic taxa for which genomic records were available (i.e., *Proteromonas lacertae, Mastigamoeba balamuthi, Brevimastigomonas motovehiculus, Reticulomyxa filosa,* and all the *Blastocystis* subtype genomes) indicating that these are in fact eukaryotic sequences and not prokaryotic contaminants (*Figure 2—figure supplement 1*). The *Proteromonas* and *Blastocystis rquA* gene sequences showed conservation of intron position and size. But none of the other eukaryotic *rquA* gene sequences (for which genomic sequence was available) shared intron positions. We did identify a potential contaminating sequence in the mollusk *Aplysia* (Bioproject:PRJNA77701, Accession number:GBCZ01101516). This homolog was 89% identical to the *Neoparamoeba* amino acid sequence and could derive from a *Neoparamoeba*-related parasite present in the *Aplysia* tissue. Indeed, we were able to identify small subunit sequence (GBCZ01078303.1) that is 93% identical to *Neoparamoeba aestuarina* in this sequencing project. For this reason, we excluded the latter from all subsequent analyses. Despite the presence of RQ in some animals (e.g., *A. suum* and *C. elegans*) we were unable to detect homologues of *rquA* in these genomes.

### Phylogenetic analysis of bacterial and eukaryotic RquA homologs

Preliminary phylogenetic analyses showed that prokaryotic and eukaryotic RquA homologs formed a maximally supported clade that emerged from within a group of bacterial ubiquinone/menaquinone biosynthesis C-methyltransferase proteins (*Supplementary file 2*, tree 1). To explore the evolutionary history of this protein, the dataset was reduced to include only the RquA sequences for subsequent analyses (*Figure 2*, *Supplementary file 2*, tree 2). The deepest split in the RquA clade as shown is between two distinct groups of homologs (here referred to as Group A and B) composed of prokaryotic and eukaryotic sequences of mixed taxonomic affinities (*Figure 2*, *Supplementary file 2*, tree 2). This split receives maximal support in both ML and Bayesian analyses in the unrooted RquA tree (*Figure 2*, *Supplementary file 2*, tree 2), as well as the larger analysis (*Supplementary file 2*, tree 1).

The Group A clade comprises sequences from alphaproteobacteria, betaproteobacteria and metagenome samples, as well as five distinct eukaryote lineages (the breviate *Pygsuia*, the stramenopiles *Blastocystis* and *Proteromonas*, three amoebozoans *Mastigamoeba, Copromyxa and a Neoparamoeba + Paramoeba group,* euglenids and the rhizarian *Brevimastigamonas*). Group B contains alpha-, beta- and gammaproteobacterial homologs, candidate phylum radiation (CPR) bacterial sequences and four independent lineages of eukaryotes (the opisthokont *Monosiga ovata*, a ciliate clade, a diatom group and a rhizarian amoebae group). Although some branches within Group A and Group B subtrees were weakly supported by bootstrap analysis and posterior probabilities, other features appeared somewhat more robust.

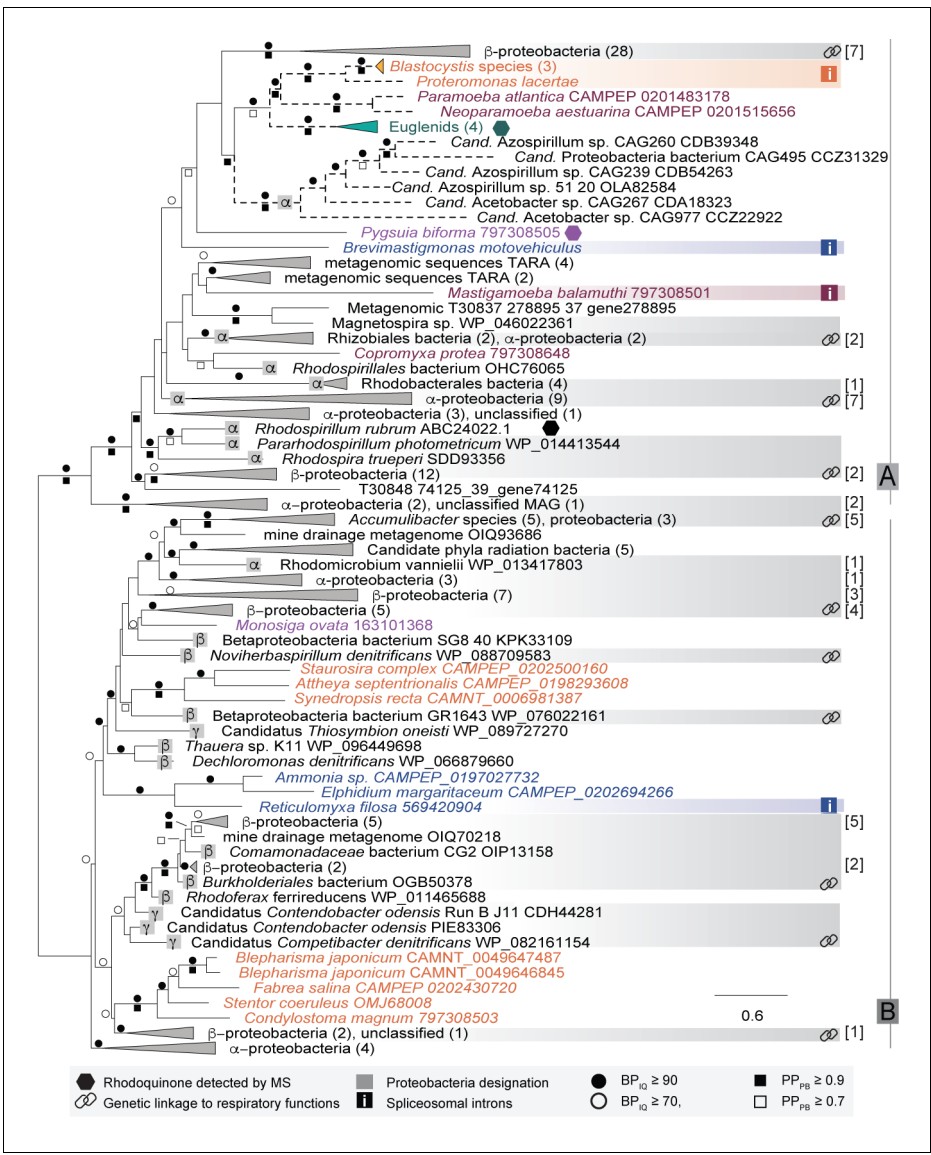

**Figure 2.** Maximum-likelihood phylogeny of RquA proteins constructed from an alignment of homologs from 166 organisms and 197 aligned sites. Eukaryotic proteins are coloured based on their phylogenetic affiliations: Obazoa (purple), Stramenopiles-Alveolata (orange), Excavata (green), Amoebozoa (magenta) and Rhizaria (blue). Hexagons represent taxa where RQ has been detected experimentally. Proteobacterial designations (α,β,γ) are indicated in the grey squares. Genetic linkage of *rquA* and genes related to respiratory function (complex I-IV, cytochrome c metabolism, or heme metabolism) are shown with chain-links and detailed in ***Supplementary file 1***. When these indications are in a collapsed node, the number of genomes showing linkage are shown in brackets. Bootstrap values (or posterior probability) greater than 70 (0.7) and 90 (0.9) are shown with open circles (squares) or closed circles (squares) respectively. The presence of spliceosomal introns in the eukaryotic sequences are indicated with 'i' in a box. Dashed branches were made shorter by 50% to facilitate visualization.

DOI: https://doi.org/10.7554/eLife.34292.003

The following figure supplements are available for figure 2:

**Figure supplement 1.** Intron positions in eukaryotic *rquA* gene sequences.

DOI: https://doi.org/10.7554/eLife.34292.004

**Figure supplement 2.** Phyletic distribution of RQUA among alphaproteobacteria superimposed on the alphaproteobacterial species tree.

DOI: https://doi.org/10.7554/eLife.34292.005

**Figure supplement 3.** RquA homologues lack critical S-adenosyl methionine (SAM) binding site.

DOI: https://doi.org/10.7554/eLife.34292.006

In Group A, *Blastocystis* + *Proteromonas* (gut stramenopile protists) grouped strongly with the parasitic amoebozoan *Neoparamoeba* + *Paramoeba* clade and the euglenids, despite the fact that none of these lineages are closely related in the tree of eukaryotes (i.e. they branch from within the three fundamental eukaryotic 'mega-groups' of Diaphoretickes, Amorphea and Excavata respectively). A weakly supported sister group to this eukaryotic subtree was formed by a clade of homologs from six metagenome assembled genomes (MAGs) from a human gut metagenome study (*Nielsen et al., 2014*). These appear to be a subgroup of the Rhodospirillales alphaproteobacteria (see below and *Figure 2—figure supplement 2*), that we henceforth refer to as 'MAG Azo/Aceto' alphaproteobacteria. All other eukaryotes in Group A branch in distinct positions in the A-subtree, with very poor support in the backbone of the tree. The breviate *Pygsuia*, the rhizarian *Brevimastigamonas* and the amoebozoans *Mastigamoeba* and *Copromyxa* were also weakly excluded from the above eukaryote + 'MAG Azo/Aceto' alphaproteobacteria grouping; each of these instead branched independently amongst bacterial groups in Group A. Group B was similarly poorly resolved, although a number of branches separating the eukaryote groups from each other did receive ultrafast bootstrap support of >90% (*Figure 2*).

We tested if several phylogenetic hypotheses could be significantly rejected by the data using approximately unbiased (AU) topology tests (*Shimodaira, 2002*) (*Table 1*, *Supplementary file 2*, trees 3–21). The first set of tests focused on the separate branching positions of eukaryotes in the overall RquA tree. A topology in which Group A and B eukaryotes were constrained to form a single clade in the tree to the exclusion of the bacterial sequences was strongly rejected (p-value = $3 \times 10^{-37}$, *Table 1*, *Supplementary file 2*, tree 2). We further tested topologies constraining the monophyly of each of the major lineages of eukaryotes coloured in *Figure 2* including: stramenopiles + alveolates (SA), stramenopiles + alveolates + rhizarians (SAR), opisthokonts + breviates, amoebozoans, Amorphea, Rhizaria, and Diaphoretickes. All topologies, except monophyletic amoebozoans (found only in Group A) were rejected with p-values < 0.05 (*Table 1*, *Supplementary file 2* trees 2–15). Tests of relationships within Group A or within Group B separately revealed that monophyletic eukaryotes in each subtree could not be rejected if each were constrained one at a time (p-value = 0.253 for Group A eukaryote monophyly and p-value = 0.179 for Group B eukaryote monophyly).

Given that *rquA* is found in many different alphaproteobacteria, one possible explanation for the origin of these genes in these protists is that they were introduced into eukaryotes by the endosymbiotic alphaproteobacterial progenitor of mitochondria. A number topology tests were conducted to address this possibility. We first tested the monophyly of alphaproteobacteria as a whole (*Supplementary file 2*, tree 18) and found this was firmly rejected (p-value = $5 \times 10^{-34}$), confirming our observation that alphaproteobacteria were phylogenetically interspersed throughout the RquA tree. We then tested: (i) the monophyly of eukaryotes + all alphaproteobacteria, (ii) the monophyly of all Group A eukaryotes + all Group A alphaproteobacteria, and the monophyly of all Group B eukaryotes + all Group B alphaproteobacteria. All of these topologies were rejected with p-values < 0.01 (*Supplementary file 2*, trees 18–21). However, a topology containing a clade of all Group A eukaryotes + 'MAG Azo/Aceto' alphaproteobacteria was not rejected (p-value = 0.227; *Supplementary file 2*, tree 17).

To improve resolution of the RquA phylogeny by including more aligned sites, we analyzed Groups A and B independently (*Supplementary file 3*, Tree 1; *Supplementary file 4*, Tree 1). As in the full dataset analysis, all eukaryote lineages in each of Group A and B did not form single monophyletic groups and showed similar branching patterns to the full analysis. However, tests did not reject topologies constraining eukaryotes to form higher-order clades within each group (*Table 1*). Like in the full data set analysis, the Group A eukaryotes + 'MAG Azo/Aceto' alphaproteobacteria topology was not rejected, but the Group A eukaryotes + all Group A alphaproteobacteria (including the MAGs) topology was strongly rejected (p-value = $3 \times 10^{-59}$). For Group B there were no alphaproteobacterial lineages particularly close to the eukaryotes in the phylogeny; eukaryotes tend to group with disparate betaproteobacteria in different parts of Group B). The test of a topology containing a clade of Group B eukaryotes + all Group B alphaproteobacteria resulted in strong rejection (p-value = $2 \times 10^{-75}$).

**Table 1.** Approximate unbiased topology tests for RquA analyses.

| Monophyly Tested | Tree file | CONSEL p-AU[a] |
|---|---|---|
| *Group A and Group B* | | |
| Maximum likelihood tree | *Figure 2*; *Supplementary file 2* - tree 2 | *0.743* |
| Group A eukaryotes + Group B eukaryotes | *Supplementary file 2* - tree 3 | 3.00E-37*** |
| Group A1 eukaryotes: *Blastocystis, Proteromonas,* Neoparamoebids, Euglenids, *Pygsuia* | *Supplementary file 2* - tree 4 | *0.622* |
| Group A1 eukaryotes + *Brevimastigamonas* | *Supplementary file 2* - tree 5 | *0.46* |
| Group A1 eukaryotes + *Brevimastigamonas* + *Mastigamoeba* | *Supplementary file 2* - tree 6 | *0.294* |
| Group A eukaryotes | *Supplementary file 2* - tree 7 | *0.253* |
| Group B eukaryotes | *Supplementary file 2* - tree 8 | *0.179* |
| Obazoa (*Pygsuia* + *Monosiga*) | *Supplementary file 2* - tree 9 | 0.002** |
| Amoerphea (Obazoa + Amoebozoa) | *Supplementary file 2* - tree 10 | 1.00E-32*** |
| Amoebozoa | *Supplementary file 2* - tree 11 | *0.206* |
| Stramenopiles + Alveolates | *Supplementary file 2* - tree 12 | 0.004** |
| Stramenopiles + Alveolates + Rhizaria (SAR) | *Supplementary file 2* - tree 13 | 0.034* |
| Diaphoretickes (SAR + Euglenids) | *Supplementary file 2* - tree 14 | 0.018* |
| Rhizaria | *Supplementary file 2* - tree 15 | 1.00E-60*** |
| Eukaryotes + MAG alphaproteobacteria | *Supplementary file 2* - tree 16 | 2.00E-41*** |
| Group A eukaryotes + MAG alphaproteobacteria | *Supplementary file 2* - tree 17 | *0.227* |
| Alphaproteobacteria | *Supplementary file 2* - tree 18 | 5.00E-34*** |
| Eukaryotes + alphaproteobacteria | *Supplementary file 2* - tree 19 | 8.00E-43*** |
| Group A eukaryotes + Group A alphaproteobacteria | *Supplementary file 2* - tree 20 | 3.00E-31*** |
| Group B eukaryotes + Group B alphaproteobacteria | *Supplementary file 2* - tree 21 | 4.00E-05*** |
| *Group A* | | |
| Maximum likelihood tree | *Supplementary file 3* - tree 1 | *0.892* |
| Eukaryotes | *Supplementary file 3* - tree 2 | *0.225* |
| Amoebozoa | *Supplementary file 3* - tree 3 | *0.315* |
| *Pygsuia* + Amoebozoa (Amorphea) | *Supplementary file 3* - tree 4 | *0.22* |
| Eukaryotes + alphaproteobacteria | *Supplementary file 3* - tree 5 | 3.00E-59*** |
| Eukaryotes + MAG alphaproteobacteria | *Supplementary file 3* - tree 6 | *0.226* |
| *Group B* | | |
| Maximum likelihood tree | *Supplementary file 4* - tree 1 | *0.827* |
| Eukaryotes | *Supplementary file 4* - tree 2 | *0.081* |
| Stramenopiles + Alveolates | *Supplementary file 4* - tree 3 | *0.287* |
| SAR | *Supplementary file 4* - tree 4 | *0.281* |
| Eukaryotes + alphaproteobacteria | *Supplementary file 4* - tree 5 | 2.00E-75*** |

[a]italicized values indicate topologies that could not be rejected ($p < 0.05$).

\* $0.05 > p > 0.01$

\*\* $0.01 > p > 0.001$

\*\*\* $p < 0.001$

DOI: https://doi.org/10.7554/eLife.34292.007
The following source data available for  Table 1:
Source data 1. Topology test output from CONSEL.
Trees 1-20 represent trees 2-21 from *Supplementary file 2*; trees 21-120 represent 100 bootstrap trees from the maximum likelihood analysis. Relevant column headers: Obs, observed log-likelihood value; au, topology test p-value; np, bootstrap probability. Details on the column headers can be found at http://stat.sys.i.kyoto-u.ac.jp/prog/consel/quick.html
DOI: https://doi.org/10.7554/eLife.34292.008
Source data 2. Topology test output from CONSEL.
Trees 1-6 represent trees1-6 from *Supplementary file 3*; trees 7-106 represent 100 bootstrap trees from the maximum likelihood analysis. Relevant column headers: Obs, observed log-likelihood value; au, topology test p-value; np, bootstrap probability. Details on the column headers can be found at http://stat.sys.i.kyoto-u.ac.jp/prog/consel/quick.html
DOI: https://doi.org/10.7554/eLife.34292.009
Source data 3. Topology test output from CONSEL.
Trees 1-5 represent trees 1-5 from *Supplementary file 4*; trees 6-105 represent 100 bootstrap trees from the maximum likelihood analysis. Relevant column headers: Obs, observed log-likelihood value; au, topology test p-value; np, bootstrap probability. Details on the column headers can be found at http://stat.sys.i.kyoto-u.ac.jp/prog/consel/quick.html
DOI: https://doi.org/10.7554/eLife.34292.010

## The distribution of *rquA* amongst alphaproteobacteria

To further investigate whether Group A or Group B eukaryotic *rquA* homologs originated from the alphaproteobacterial mitochondrial endosymbiont, we investigated the representation and phylogenetic distribution of available *rquA*-containing alphaproteobacterial genomes. We assembled a phylogenomic matrix of 200 conserved 'core' proteins from the alphaproteobacteria (*Wang and Wu, 2013*) to place the *rquA*-containing alphaproteobacterial taxa from our analyses within the context of a representative alphaproteobacterial species tree. Note that the matrix was assembled in such a way to always represent *rquA*-containing taxa, with the remainder of alphaproteobacterial taxa subselected from available genomes to maximize diversity. A maximum likelihood phylogeny estimated from this matrix shows that *rquA*-encoding alphaproteobacterial genomes are patchily distributed, emerging as numerous isolated groups within several orders of the alphaproteobacteria (*Figure 2—figure supplement 2*). Group A- and Group B-containing taxa are interspersed. Genomes encoding the gene are extremely rare within the alphaproteobacterial orders. Five alphaproteobacterial orders (Rickettsiales, Holosporales, Pelagibacterales, Sphingomonadales, and Caulobacterales) had no *rquA*-containing taxa (out of a total of 455 genomes examined). The remaining three orders had *rquA*-containing taxa in 7 out of 514 (Rhizobiales), 6 out of 513 (Rhodobacterales) and 21 out of 217 (Rhodospirillales). The *rquA*-encoding 'MAG Azo/Aceto' alphaproteobacteria branch from within the Rhodospirillales as sister to another *rquA*-containing group. It is notable that these latter taxa branch separately the RquA phylogeny (*Figure 2*).

## Genomic context and primary sequence analysis of RquA

To determine if *rquA* is genetically linked to other potential RQ biosynthesis genes, the genomic context of *rquA* in the various bacterial genomes was investigated (chain link icons in *Figure 2*, *Supplementary file 1*). RquA genes do not appear to be located near genes encoding other hypothetical proteins or candidate quinone biosynthesis enzymes. However, in many of the bacterial genomes, *rquA* is encoded close (i.e., within 15 genes) to the genes encoding respiratory complexes (e.g., complex I, II, or III), respiration-associated functions (e.g., cytochrome, ubiquinone, and heme biosynthesis) (*Figure 2*, *Supplementary file 1*), and/or anaerobiosis-associated proteins (e.g., [NiFe]-hydrogenase, nitrate reductase, [FeFe]-hydrogenase). The genetic proximity we observed between *rquA* and genes encoding CII and other respiratory subunits suggests they could be transcriptionally linked in an operon in these bacteria. Furthermore, in 12 of these bacteria, there is second *frd/sdh* operon located elsewhere in the genome suggesting that a different complex might be expressed under low versus high oxygen conditions, as was shown for *E. coli* (*Jones and Gunsalus, 1985*) and *Ascaris suum* (*Iwata et al., 2008*).

Examination of the primary sequence of the bacterial and eukaryotic RquA homologs suggest that these proteins belong to a family of Class I S-adenosyl methionine (SAM) methyltransferases, which includes the UQ methyltransferases UbiE (pfam08241). A survey of methyltransferases identified four distinct motifs common to most class I SAM methyltransferases (motif I, motif post-I, motif

II and motif post-II) that are crucial for protein folding and SAM binding (*Petrossian and Clarke, 2009*). While three of these motifs are conserved in RquA sequences, the SAM-binding motif I is different when compared to other Class I SAM-methyltransferases (*Figure 2—figure supplement 3*). Like the bacterial RquA sequences discussed by Lonjers and colleagues (*Lonjers et al., 2012*), the eukaryotic RquA sequences have substitutions in key SAM binding sites (*Figure 2—figure supplement 3*).

## The distribution of quinone-utilizing enzymes in eukaryotes

RQ is known to function as an electron carrier between complex I, complex II and other UQ-utilizing enzymes such as electron transferring flavoprotein dehydrogenase (*Ma et al., 1993*). If RquA is in fact synthesizing RQ in the eukaryotes presented here, then these organisms must encode at least two types of RQ-utilizing enzymes: those that reduce RQ and others that reoxidize it. To test this hypothesis, we searched for genes encoding the following quinone-utilizing enzymes that could possibly interact with RQ in the *rquA*-containing eukaryotic genomes and transcriptomes: the respiratory complexes (CI, CIII); quinone biosynthesis enzymes, COQ1-7; alternative oxidase (AOX); dihydroorotate dehydrogenase (DHOD); electron transferring flavoprotein system (ETF) (made up of ETFα and β; ETF dehydrogenase, ETFDH); glycerol-3-phosphate dehydrogenase (G3PDH); and sulfide:quinone oxidoreductase (SQO). All *rquA*-containing eukaryotes including the MRO-containing protists (e.g., *Pygsuia*, *Mastigamoeba* and *Blastocystis*) possess at least CII and ETF as well as up to four other quinone-utilizing complexes (*Figure 3*; *Supplementary file 1*). Whereas most aerobic model system eukaryotes possess the majority of these Q-utilizing systems in their mitochondria (*Marreiros et al.,*

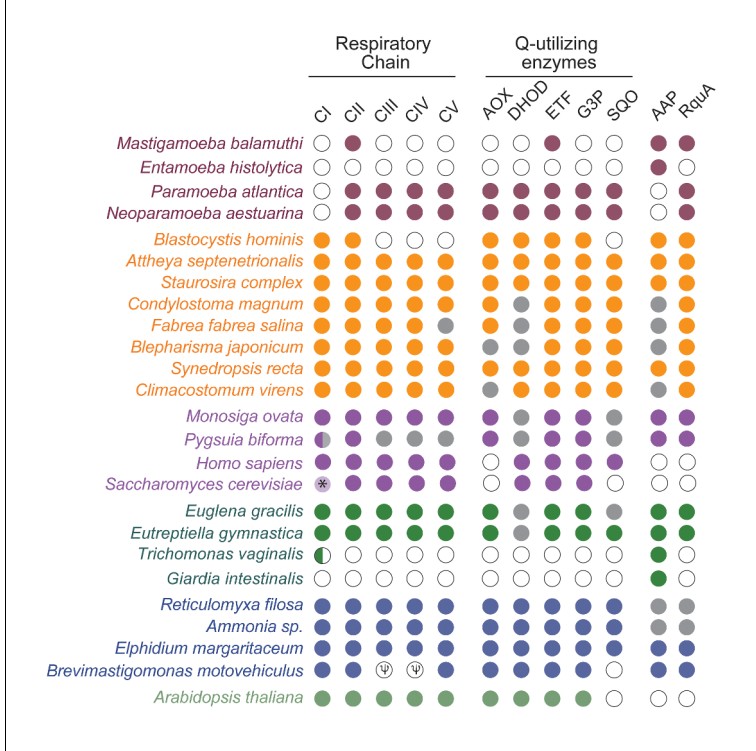

**Figure 3.** RquA co-occurs with quinone-utilizing enzymes. Eukaryotic genomes and transcriptomes were surveyed for homologs of respiratory chain components (Complexes I-V, CI-CV), alternative oxidase (AOX), dihydroorotate dehydrogenase (DHOH), electron-transferring flavoprotein system (ETF), glycerol-3-phosphate dehydrogenase (G3P), sulfite:quinone oxidoreductase (SQO), RquA, and one or more anaerobiosis-associated protein (AAP; detailed in *Supplementary file 1*). Grey and white circles indicate that homologs were not detected in transcriptome and genome sequence data respectively. Half circle in CI for *Pygsuia biforma* and *Trichomonas vaginalis* indicates only two subunits (NUOE and NUOF) were identified. 'ψ' indicates pseudogenes.
DOI: https://doi.org/10.7554/eLife.34292.011

*2016*), the anaerobic protists that completely lack them, such as *Giardia intestinalis*, *Trichomonas vaginalis* and *Entamoeba histolytica*, also lack *rquA* (*Figure 3*).

## Subcellular localization of RquA and RQ production in *Pygsuia biforma*

In many eukaryotes, UQ biosynthesis occurs in the mitochondrion (*Tran and Clarke, 2007*), and since UQ is a known precursor to RQ in *R. rubrum* (*Brajcich et al., 2010*) it is possible that RQ biosynthesis (and therefore RquA) would also be localized to mitochondria (or MROs). To test if eukaryotic RquA homologs could function in mitochondria, we evaluated the presence of N-terminal mitochondrial targeting signals (MTS) using publically available software tools (*Claros and Vincens, 1996*; *Emanuelsson et al., 2000*; *Fukasawa et al., 2015*). Each eukaryotic sequence that was full-length had a predicted MTS by at least two of these predictors (*Supplementary file 1*). To experimentally test these predictions, we used spinning-disk confocal microscopy to localize RquA by indirect immunofluorescence microscopy in the anaerobic protist *Pygsuia biforma*. The immunofluorescence detection of RquA using antibodies raised against *Pygsuia* RquA produced puncta that co-localized with the mitochondrion-reactive stain MitoTracker in *Pygsuia* (*Figure 4*, *Figure 4—figure supplement 1*), which indicates this protein functions in the MROs of this organism.

To test whether the presence of *rquA* in *Pygsuia* correlates with the production of RQ, we examined its quinone content by selected-reaction monitoring (SRM) mass spectrometry. We first determined the high-performance liquid chromatography retention times and fragmentation patterns of different isoprenylated states of rhodoquinone and ubiquinone (e.g., $RQ_3$, $RQ_{8-10}$, and $UQ_{8-10}$; *Figure 4*, *Figure 4—figure supplement 2*). With these parameters determined, we investigated the lipid components of *Pygsuia biforma* (grown with a mixed culture of prokaryotes) and the prokaryote community (grown without *Pygsuia*). In the first community, but not the second, we observed three species that correlated with the retention times and fragmentation profiles of $RQ_{8-10}$ (*Figure 4*, *Figure 4—figure supplement 3*) suggesting that *Pygsuia biforma* synthesizes RQ.

## Discussion

### Origin of RquA and rhodoquinone biosynthesis

The ability to produce RQ and use fumarate as a terminal electron acceptor in the mitochondrial ETC appears to be an adaptation to low-oxygen conditions that is present in many independent eukaryote lineages. Our examinations of publicly available genome and transcriptome sequence data, revealed genes encoding the RQ biosynthesis protein RquA in multiple, and yet sparsely distributed, lineages of eukaryotes and bacteria. Our preliminary phylogenetic analysis revealed that all RquA homologs evolved from within a larger clade of proteobacterial Class I SAM-dependent methyltransferases. The closest homologs of RquA are, in fact, members of the UQ biosynthesis pathway. Interestingly, previous reports have demonstrated that RQ is synthesized from UQ, and not a precursor of UQ in *R. rubrum* (*Brajcich et al., 2010*). Therefore, it seems plausible that RquA originally evolved from a family of proteins already capable of binding UQ or similar molecules and eventually gained a new enzymatic activity to function in RQ biosynthesis.

Within the eukaryote domain, interactions between redox proteins and RQ have only been investigated in detail in *Ascaris suum*, *Caenorhabditis elegans*, and *Euglena gracilis* (*Kita et al., 1988*; *Takamiya et al., 1999*; *Hoffmeister et al., 2004*; *Castro-Guerrero et al., 2005*). While *rquA* is found in the transcriptome of *E. gracilis*, we could not detect homologs in the genomes of *A. suum*, *C. elegans* or closely related helminths and nematodes. This suggests that the RQ biosynthesis pathway of these metazoans may not be related to the RquA-based system and that these organisms have convergently evolved the capacity to synthesize and utilize RQ in adaptation to hypoxia.

### Eukaryotic RquA likely functions in mitochondria

The terminal steps of UQ biosynthesis typically occur in mitochondria (*Tran and Clarke, 2007*). Assuming that UQ is a precursor of RQ (*Brajcich et al., 2010*), RQ biosynthesis could also occur in the mitochondrion or MROs of eukaryotes. However, not all of the eukaryotes shown to have *rquA* have the genetic repertoire necessary for endogenous UQ biosynthesis (*Supplementary file 1*). While the exact mechanisms of how exogenous UQ is transported into model eukaryotic cells is unknown (*Padilla-López et al., 2009*), studies have demonstrated that mouse cells preferentially

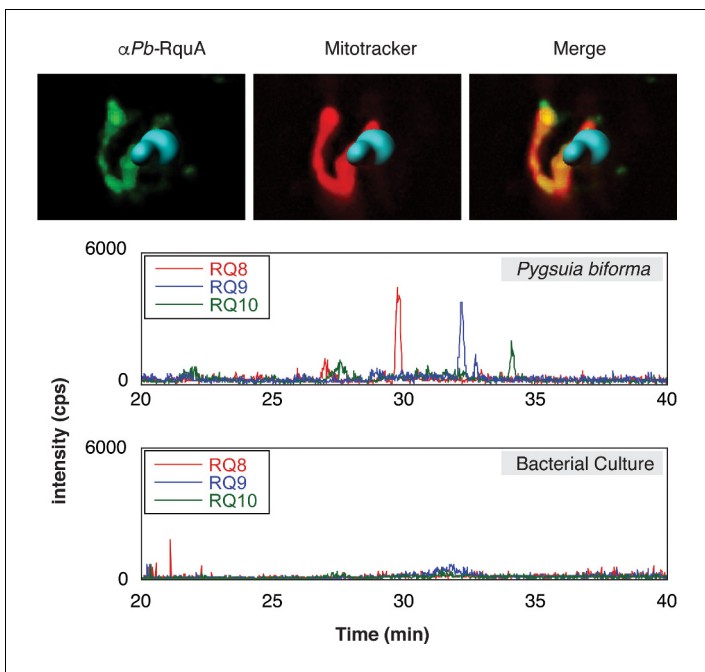

**Figure 4.** Subcellular localization of RquA and rhodoquinone production in *Pygsuia biforma*. (**A**) Antibodies raised against RquA (green) colocalized with MitoTracker (red). Confocal slices (0.3 μm) were deconvoluted (using a constrained interative algorithm) and combined to render a 3D image. DAPI stained nuclei (blue) were volume rendered in Imaris for clarity. (**B**) Lipid extracts were separated by liquid chromatography and analyzed with selected-reaction monitoring mass spectrometry. Rhodoquinone species eluted from the column in roughly 3 min intervals as chain length increases (RQ8-10). Diagnostic product ions corresponding to the rhodoquinone head group (182.1 m/z) following fragmentation of parent ions were detected.

DOI: https://doi.org/10.7554/eLife.34292.012

The following figure supplements are available for figure 4:

**Figure supplement 1.** Antibodies directed against Pygsuia RquA (*Pb*RquA) recognize purified recombinant *Pb*RquA by western blot analysis.

DOI: https://doi.org/10.7554/eLife.34292.013

**Figure supplement 2.** Spectra represent the elution times of detection of the rhodoquinone head group (182.1 m/z; left) and ubiquinone head group (197.1 m/z) following fragmentation of parent ions of different isoprenyl chain lengths as indicated.

DOI: https://doi.org/10.7554/eLife.34292.014

**Figure supplement 3.** Lipid extracts were separated by high-performance liquid chromatography and analyzed with mixed-reaction monitoring mass spectrometry from *Pygsuia* mixed culture, *Pygsuia* 'food' bacteria culture, glassware, and *Rhodospirillum* rubrum as indicated.

DOI: https://doi.org/10.7554/eLife.34292.015

incorporate exogenously supplied UQ into mitochondrial membranes (*Lapointe et al., 2012*). We propose that these heterotrophic protists obtain UQ from their bacterial prey, and convert this UQ to RQ by RquA in their MROs.

All of the eukaryotic homologs of RquA with complete N-termini have predicted mitochondrial targeting signals suggesting mitochondrial localization (*Supplementary file 1*). These predictions were experimentally validated for *Pygsuia biforma* using immunofluorescence microscopy; RquA localizes to the MRO in these cells (*Figure 4*). In model organisms, most of the enzymes that use UQ as an electron carrier are also found in mitochondria, such as CI, CII, alternative oxidase, and the electron transferring flavoprotein/dehydrogenase system (*Wang and Hekimi, 2016*). Interestingly, every eukaryotic organism that encodes RquA (including anaerobes with a reduced electron transport chain) also encodes CII and at least one other UQ-utilizing enzyme (minimally CII and ETF; *Figure 3*, *Supplementary file 1*). Since some of these proteins have been shown to interact with RQ in *Ascaris suum* (*Ma et al., 1993*; *Iwata et al., 2008*), it seems likely that the UQ-utilizing complexes of

RquA-containing eukaryotes, are also capable of using RQ as a cofactor under certain conditions (i. e., anoxia).

## The evolutionary history of RquA in bacteria

Our detailed analyses of the RquA phylogeny revealed that the relationships between sequences is incongruent with expected organismal relationships for both the eukaryotes and bacteria. RquA is extremely rare and patchily distributed over the backbone phylogeny of the groups of organisms in which it is found. Amongst bacteria, *rquA* is found in only 36 genomes of 2040 alphaproteobacteria, 59 out of 946 genomes of betaproteobacteria, and 2 out of 2493 genomes of gammaproteobacteria present in Genbank (September 2017). If one were to explain the bacterial *rquA* distribution by vertical inheritance alone, the gene would have had to have been present in the common ancestor of alpha-, beta- and gammaproteobacteria, and been lost potentially hundreds of times independently in the various members of the phyla that do not encode *rquA*. Our investigation of the phylogenetic distribution of *rquA* amongst alphaproteobacterial genomes specifically confirms this interpretation; the gene is extremely patchily distributed amongst orders of alphaproteobacteria and is completely absent from five of these orders. This is consistent with the rejection of alphaproteobacterial monophyly in the RquA phylogeny (*Table 1*). It is therefore extremely difficult to determine in which group of bacteria *rquA* might have originated. The 'scrambled' phylogenetic affinities amongst bacterial *rquA* homologs is a relatively common phenomenon for genes encoding enzymes in bacteria because of the high rate of LGT in prokaryotic genome evolution (*Eisen, 2000*; *Ochman et al., 2000*; *Kunin and Ouzounis, 2003*).

## The evolutionary history of RquA in eukaryotes

The non-monophyly of all eukaryote RquA homologs coupled with the rarity and patchy distribution of the gene amongst eukaryotes (*Figure 2*) argues against a simple vertical inheritance explanation. Our topology tests clearly indicate that a single global eukaryotic *rquA* clade and any higher-order grouping of Group A and B eukaryote sequences are strongly rejected by the data. However, within Group A and Group B, the monophyly of eukaryotic homologs cannot be rejected. Therefore, at the very least, Group A and Group B *rquA*s were introduced as different genes into eukaryotic genomes. The presence of *rquA* in some alphaproteobacterial genomes makes an endosymbiotic origin of eukaryote *rquA* an attractive hypothesis. If both A- and B-types of *rquA* had an endosymbiotic origin and were strictly vertically inherited within eukaryotes, then the endosymbiotic progenitor of mitochondria must have encoded both types. This is improbable given that no bacterium or eukaryote currently known (out of 182 organisms possessing the gene) encodes both types. However, it is also possible that either Group A or Group B *rquA*s in eukaryotes had an endosymbiotic origin and the other type was later re-acquired by bacteria-to-eukaryote LGT after loss of the ancestral endosymbiotic type. Of the two types, Group A *rquA* seems to be a more likely candidate for a mitochondrial origin since a Group A eukaryotes + 'MAG Azo/Aceto' alphaproteobacteria clade is not rejected in RquA topology tests. However, there are several problems with this scenario. First, 'MAG Azo/Aceto' alphaproteobacteria are a subgroup emerging from *within* the Rhodospirillales alphaproteobacteria (*Figure 2—figure supplement 2*) and are not recovered as the sister group of mitochondria in recent phylogenomic analyses. Such analyses suggest that mitochondria are either related to the Rickettsiales (*Wang and Wu, 2015*) and/or Pelagibacteriales (*Thrash et al., 2011*), or emerge as an independent deep branch within alphaproteobacteria (*Martijn et al., 2018* ). Since *rquA* is completely absent from Rickettsiales and Pelagibacteriales, and is very unlikely to be ancestral to alphaproteobacteria as a whole (*Figure 2—figure supplement 2*), a mitochondrial origin for eukaryotic homologs seems unlikely.

It is also improbable that either Group A or Group B *rquA* is ancestral to all eukaryotes. For example, if Group A *rquA* genes were ancestral to all eukaryotes, *many* parallel secondary loss events would have to be postulated. Considering the absence of the gene in genome sequence data of diverse eukaryotes (*Sibbald and Archibald, 2017*), the list of eukaryotic groups that would have to have lost A-type *rquA* includes multiple amoebozoan lineages except *Mastigamoeba* and neoparamoebids, lineages of stramenopiles besides *Blastocystis* and *Proteromonas* (e.g. diatoms, ochrophytes, oomycetes), lineages of Rhizaria besides *Brevimastigamonas* (e.g. *Bigelowiella* and *Reticulomyxa*), multiple Excavata lineages besides euglenids (e.g. Kinetoplastida, Metamonada,

Heterolobosea), and the common ancestors of opisthokonts, apusomonads, alveolates, haptophytes, cryptophytes and Archaeplastida. The number of secondary losses required to explain Group B *rquA* as an ancestral eukaryotic gene is similarly large. Both secondary loss lists become considerably longer if the absence of the gene from transcriptomes of diverse eukaryotes (*Sibbald and Archibald, 2017*) (*Supplementary file 1*) is considered as evidence.

Collectively these lines of evidence suggest that neither A- nor B-types of *rquA* are of mitochondrial origin and neither were present in the last eukaryotic common ancestor. Instead, these observations are more consistent with an LGT-based explanation, especially given that all of the eukaryotes that possess *rquA* genes are either anaerobes or facultative anaerobes. Although the role of LGT in eukaryotes has recently been questioned (*Ku et al., 2015*; *Martin, 2017*), there is now abundant evidence for prokaryote-to-eukaryote and eukaryote-to-eukaryote LGT (*Andersson, 2009*; *Jerlström-Hultqvist et al., 2013*; *Stairs et al., 2014*; *Karnkowska et al., 2016*; *Eme et al., 2017*; *Husnik and McCutcheon, 2018*; *Leger et al., 2018*). An LGT scenario for *rquA*, would require at least two bacteria-to-eukaryote LGTs (minimally one for Group A and one for B) followed by events of eukaryote-to-eukaryote LGT within groups A and B (*Figure 2*). The scrambled phylogenetic relationships amongst the *rquA*-containing bacterial groups in the RquA tree makes it difficult to discern potential donors of eukaryotic *rquA*s. However, for the eukaryote sequences of the A-type, the affinities of the 'MAG Azo/Aceto' alphaproteobacteria to some of the eukaryotic sequences could implicate this gut-dwelling lineage of the Rhodospirillales as donors. This is particularly intriguing since the same genomes encode other enzymes of anaerobic metabolism including [FeFe]-Hydrogenase and its maturase enzymes HydE-G (*Degli Esposti et al., 2016*; *Leger et al., 2016*). Although several of these 'MAG Azo/Aceto' alphaproteobacterial homologs of hydrogenase metabolism show affinities to some (but not all) eukaryotic homologs, others appear to be more distantly related to eukaryotes (and to each other) suggesting multiple independent origins by LGT (see supplementary figures S3-S6 from *Leger et al., 2016*). We propose that the 'MAG Azo/Aceto' alphaproteobacteria acquired the hydrogen and rhodoquinone metabolism genes from other anaerobic bacteria relatively recently in a series of LGT events.

In any case, after the initial lateral transfer of A- and B-type genes into recipient eukaryotes, the *rquA* sequences must then have acquired mitochondrial targeting sequences (MTS). The subsequent transfers of MTS-containing *rquA* genes between different eukaryotes could then have allowed them to adapt their mitochondrial ETCs to low-oxygen conditions. Additional data from more eukaryotic and prokaryotic taxa harbouring *rquA* (perhaps from metagenomic investigations of low-oxygen environments) should improve the resolution of the phylogeny and will allow the relative merits of the various evolutionary origin scenarios to be re-assessed in future.

## RquA function and the 'transferability' of RQ biosynthesis between organisms

The role and enzymatic activity of RquA in this reaction is currently unclear. *Lonjers et al. (2012)* outline three hypotheses for the role of RquA in RQ biosynthesis. The first hypothesis is that RquA could serve a regulatory role in the expression of currently unknown RQ biosynthesis proteins. Since the vast majority of mitochondrial proteins (and all of those involved in quinone biosynthesis) are encoded by nuclear genes and we have shown that RquA functions in mitochondria, it is unlikely that RquA has a role in regulating the expression of such genes in the nucleus (although it could participate in post-transcriptional, or post-translational regulation).

A second hypothesis is that UQ and RQ are synthesized by different multi-enzyme complexes in *R. rubrum* that share *some* components. Therefore, the removal of one of these components (RquA) results in a non-functional complex in their *rquA* knockout experiments. This is based on the observation that some components of the yeast UQ biosynthesis complex are necessary for stabilization of the complex independent of their catalytic role (*Baba et al., 2004*). If this were the case for RQ biosynthesis, one would expect all organisms that encoded RquA would also have to encode UQ biosynthesis proteins. However, many of the eukaryotes that encode RquA do not encode any components of the UQ biosynthesis pathway (*Supplementary file 1*). These organisms either synthesize UQ by an unknown mechanism, or rely on exogenous sources of UQ as discussed above. It is therefore very improbable that RquA serves to stabilize a UQ/RQ biosynthesis complex.

The final hypothesis is that RquA is directly involved in catalyzing the conversion of UQ to RQ. Since RquA is related to the quinone biosynthesis enzyme UbiE/Coq5 (*Supplementary file 2*, tree

1), RquA very likely can bind to UQ-like molecules. Although it is tempting to suggest that RquA directly converts UQ to RQ in one step, no enzyme is known that catalyzes a one-step amino-transfer reaction with a methoxy leaving group. It seems more plausible that either: (i) RquA alone catalyzes a multi-step reaction sequence, or, (ii) is one enzyme in a multi-step pathway involving at least a demethylation step followed by an amino-transfer reaction. We were unable to find any other enzyme with the same restricted phylogenetic distribution as *rquA* amongst bacterial or eukaryotic homologs using phylogenetic profiling (i.e., assessing co-presence of genes within *rquA*-containing but not *rquA*-lacking organisms). Therefore, it seems likely that, if option (ii) were correct, another broadly-conserved enzyme with a different main biochemical function in bacteria and in mitochondria could, as a moonlighting side-reaction, catalyze a step of RQ biosynthesis. Under either of these scenarios, then, the lateral transfer of the *rquA* gene could, in principle, confer on the recipient the ability to convert UQ to RQ. Clearly, further experimental work will be needed to test these hypotheses.

## Remodeling of the electron tansport chain

Some of us have previously advanced a hypothetical scenario that proposes evolutionary steps by which the diverse types of anaerobic mitochondrion-related organelles might have evolved in protistan lineages adapted to transient or permanent hypoxia from predominantly aerobic ancestors (*Stairs et al., 2015*). The acquisition of *rquA* by LGT as described herein, was a key early step in these transitions; the selective benefit of this acquisition seems relatively clear. For example, assuming that the ancestral mitochondrial metabolism of many aerobic protistan groups resembled that of modern model system eukaryotes (*Figure 5*, 'aerobic mitochondria'), it is likely that they would occasionally encounter transient low-oxygen conditions regardless of whether they are in marine, freshwater, or terrestrial environments. During transient exposure to anoxia, it is known from model systems that CIII and CIV fail to function efficiently and are downregulated (*Vijayasarathy et al., 2003*; *Fukuda et al., 2007*). Furthermore, under hypoxic conditions (e.g., in ischemia-reperfusion injury or solid tumours), shifts in mitochondrial metabolism occur: reduced UQ ($UQH_2$) builds up because of the lack of CIII/CIV activity, the NADH/NAD + ratios increase as CI throughput is repressed, and part of the TCA cycle reverses (*Tomitsuka et al., 2010*; *Chouchani et al., 2014*). Under these conditions, malate is imported from the cytosol (or produced in mitochondria) and is converted to fumarate by fumarase. At sufficiently high $UQH_2$ and fumarate concentrations, CII will function in fumarate reduction to regenerate UQ (*Ackrell et al., 1993*) and succinate. This 'NADH: fumarate reductase system' allows CI to continue to function to pump protons and oxidize NADH (*Tomitsuka et al., 2010*). Under these conditions, however, it is known that CI and CII produce high levels of toxic reactive oxygen species (ROS) *Tomitsuka et al., 2010*; *Chouchani et al., 2014*. Thus, if an organism frequently encountering these low-oxygen conditions were to acquire *rquA* by LGT and express the gene, the organism would able to synthesize and utilize low electron potential RQ as an electron carrier. RQ would greatly enhance the efficiency of the LGT recipient's NADH:fumarate reductase system leading to an increase in proton pumping of CI, restoration of ATP synthesis and decreased ROS production.

The foregoing scenario is likely why *rquA* was retained after acquisition by protists with canonical mitochondria that can function aerobically and anaerobically such as *Euglena* (*Hoffmeister et al., 2004*; *Castro-Guerrero et al., 2005*) and a number of the ciliates (*Figure 2*). We propose that the acquisition of *rquA* is likely one of the first steps in the evolution of anaerobic MROs found in protists such as *Brevimastigamonas, Blastocystis, Mastigamoeba,* and *Pygsuia*. In these lineages, there has been further reductive evolution in their ETC, as they have adapted to thrive predominantly in hypoxic environments. For example, CIII and CIV of the electron transport chain of the recently described rhizarian anaerobe *Brevimastigamonas motovehiculus* appears to be degenerating with components missing or becoming pseudogenes (*Gawryluk et al., 2016*). The parasites *Blastocystis* spp. have gone further in this reduction; they completely lack CIII-CV (*Lantsman et al., 2008*; *Stechmann et al., 2008*; *Gentekaki et al., 2017*) and therefore none of their ATP is produced by oxidative phosphorylation. The most reduced ETCs amongst organisms with RquA occur in *Pygsuia biforma* and *Mastigmoeba balamuthi* (*Figure 5*), as both completely lack CI, CIII, CIV and CV. In the absence of CI, it is unclear how these two organisms can be reducing the oxidized RQ produced by CII. Most of their quinone-binding proteins typically catalyze redox reactions with reduction potentials far greater than RQ reduction and thus would be expected to oxidize $RQH_2$ rather than do the

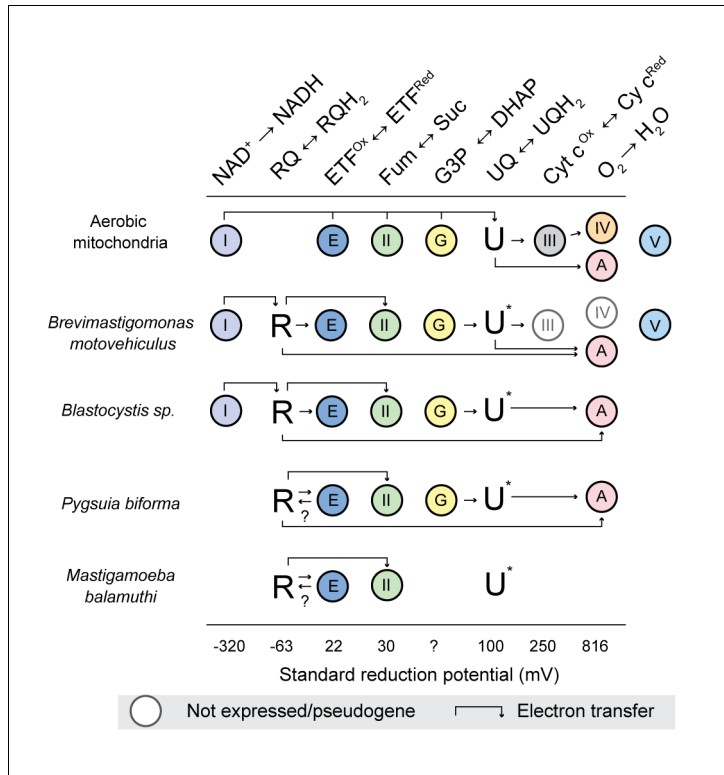

**Figure 5.** The interactions of rhodoquinone with other mitochondrial redox reactions in different eukaryotes with mitochondrion-related organelles. Standard reduction potentials for each major reaction involved in rhodoquinone (R)) and ubiquinone (U) metabolism are shown in increasing order of potential. Half reaction equations are detailed in **supplementary file 1**. The electron transfer is more favourable when passed to a species with a more positive standard reduction potential (i.e. from left to right). Abbreviations: I, Complex I; II, Complex II; III, Complex III; IV, Complex IV; V, Complex V; E, electron transferring flavoprotein dehydrogenase; G, glycerol-3-phosphate (G3P) dehydrogenase; A alternative oxidase; Fum, fumarate; Suc, succinate; DHAP, dihydroxyacetone phosphate; and Cyt c, cytrochrome c. U* Indicates that the involvement of ubiquinone is unknown, ? indicates the direction of electron transfer is unknown. Absence of a circle indicates that no homologs were detected in the organism. Genes undergoing pseudogenization are shown in white cirlces.
DOI: https://doi.org/10.7554/eLife.34292.016

opposite. One possibility is that the electron transferring flavoprotein (ETF)/ETF-dehydrogenase (ETF-DH) systems of these organisms (which usually function in fatty acid oxidation) are able to gain electrons by oxidizing reduced NADH. This could occur by electron bifurcation with butyryl-CoA dehydrogenase, as is known to occur in some bacteria (*Chowdhury et al., 2014*; *Chowdhury et al., 2015*). These bacterial ETF alpha and beta subunits can coordinate two FAD cofactors (instead of one as in mitochondrial ETF), allowing for electron bifurcation (*Roberts et al., 1996*; *Chowdhury et al., 2015*). An alternative possibility in *Pygsuia* is that NADH oxidation could be achieved by two remaining conserved subunits of CI, NuoE and NuoF (*Stairs et al., 2015*), working together with ETF/ETF-DH system. NuoE and F are thought to be involved in oxidation of NADH and ferredoxin in conjunction with [FeFe]-hydrogenase in many anaerobic protists to produce molecular hydrogen (*Stairs et al., 2015*; *Hamann et al., 2016*). In any case, if ETF could oxidize a low redox potential cofactor (e.g., NADH and/or ferredoxin), it could pass the electrons on ETF-DH (present in both *Pygsuia* and *Mastigameoba*) that could replenish the $RQH_2$ pool in these organisms (shown by arrows in *Figure 5*). Biochemical studies of the ETCs in these anaerobic protists will be critical to test these hypotheses.

## Conclusions

The gene encoding RquA, an enzyme required to synthesize RQ, was likely transferred from bacteria to various eukaryote lineages by multiple independent events of LGT, well after the establishment of mitochondria within eukaryotes. *RquA* was then subsequently transferred between some eukaryotes *via* LGT, although the directions of these transfers cannot be easily discerned. Regardless, these LGTs likely conferred the ability to biosynthesize RQ in mitochondria, making it possible for complex II to efficiently reduce fumarate to succinate, allowing the recipient eukaryotes to respire in the absence of oxygen. This is a stark example of how laterally acquired enzymes can interface with ancestral pathways in mitochondria to rapidly adapt these organelles to low oxygen conditions (*Stairs et al., 2014*; *Nývltová et al., 2015*).

# Materials and methods

## Culturing and microscopy

Cultures of *Pygsuia biforma* were maintained in American Type Culture Collection medium 802 prepared in natural seawater as described previously (*Brown et al., 2013*; *Stairs et al., 2014*). Cells were grown in 15 mL culture tubes filled with media and supplemented with *Klebsiella pneumonia.* The protein sequence of *Pygsuia* RquA was provided to Genscript for antigen design and the most antigenic peptide sequence (CGGKAVFIDYGRPST) was selected for optimal for immunization. Antibodies were generated in rabbits and affinity purified by Genscript. A dilution of 1:200 as used for immunofluorescence on *Pygsuia* cells as described previously (*Stairs et al., 2014*) . Fluorescence micrographs were deconvolved using a constrained interative algorithm in Slidebook 6 (Intelligent Imaging Innovations, Boulder, CO) and 3D images was rendered using Imaris 7.1 software (Bitplane Inc. South Windsor, CT).

## Molecular biology

The *Pygsuia rquA* gene was amplified from cDNA using primers designed with BamHI restriction enzyme recognition sites near their 5'-ends (Pb-*rquA*-forward CC<u>GGATCC</u>ATGAATTCTTTAAGAA TTAC and Pb-*rquA*-reverse CCC<u>GGATCCT</u>GCAATGCGGTGTGCAACAACC; restriction enzyme recognition sites are underlined). The amplicons were purified and cloned into the sequencing vector pCR4 (Life Technologies, Carlsbad, California) by TA-cloning. Plasmids (pCR4-*Pb-rquA*) were purified from transformed *E. coli* using the Nucleospin plasmid purification kit (Machery Nagel, Germany) and screened for correct sequence (Genewiz, South Plainfield, New Jersey). Destination plasmid pGEX-4T-1 (GE healthcare, Chicago, Illinois ) and pCR4-*Pb-rquA* were digested with BamHI (ThermoFisher, Waltham, Massachusetts). Fragments were purified using the Extract II kit (Machery Nagel) and cloned by standard protocols to generate pGEX-*Pb-rquA*.

## Heterologous expression of proteins in *E. coli* and immunoblotting

Plasmids (pGEX-Pb-rquA) were transformed into *E. coli* (strain BL21) for protein expression. Protein expression was induced by the addition of 1 mM isopropyl β-D-1-thiogalactopyranoside (Sigma, Saint Louis, Missouri) to the culture medium of exponentially growing cells and allowed to grow for an additional 4–6 hr. Proteins were isolated from *E. coli* cells: (i) induced to express the Pygsuia protein (GST-RquA) or (ii) induced to express only the GST protein or (iii) that were not induced to express protein. After protein expression, *E. coli* cells were collected by centrifugation (4000 x g, 5 min, 4°C) and lysed by French press (7000 psi). Unbroken cells and debris were removed by centrifugation (4000 x g, 5 min, 4°C). The resulting supernatant was saved for subsequent analysis and recombinant protein was isolated using glutathione-magnetic beads (ThermoFisher; GST-tag) according to the protocol of the manufacturer. Crude cell lysates and purified RquA were denatured in sample loading buffer (Sigma), boiled for 5 min, and resolved by SDS-PAGE (12%). Proteins were transferred to PVDF membranes (Turbo Blot membranes, Biorad, Hercules, California) that were then incubated in blocking buffer (5% skim milk powder, TBS, 0.5% Tween 20) for 1 hr. Anti-*Pygsuia* RquA antibodies were diluted in blocking buffer (1:500) and incubated with membranes overnight. Following three washes in TBS-tween (TBS, 0.5% Tween 20), membranes were incubated with horseradish peroxidase-conjugated goat anti-rabbit secondary antibodies in blocking buffer (1:50000, Sigma), washed in TBS-tween and incubated with enhanced chemiluminesence substrate (GE

Healthcare) and visualized using a charge-coupled-device chemiluminescence detector (Protein Simple, San Jose, California).

## Phylogenetic dataset construction and sequence analysis

For the phylogenomic analysis of alphaproteobacteria, predicted proteomes were downloaded from National Centre for Biotechnology and Information (http://www.ncbi.nlm.nih.gov/) protein database. Gene markers were identified using the Phyla-AMPHORA pipeline (*Wang and Wu, 2013*). Phya-AMPHORA identified 200 'phylum-specific' gene markers in the phylum-level bacterial phylogenetic marker database that are phylogenetically congruent for the Alphaproteobacteria.

For evolutionary analyses of RquA, eukaryotic and prokaryotic homologs of RquA, and close homologs of the UBIE family, were retrieved via BLASTP and TBLASTN (*Altschul et al., 1997*) using the *R. rubrum* RquA sequence as a query against the expressed sequence tag (EST), whole genome shotgun contigs, transcriptome shotgun assemblies and *nr* databases available at the National Centre for Biotechnology and Information (http://www.ncbi.nlm.nih.gov/). The reads for *Euglena mutabilis* (bioproject ERR351290) were assembled using DNA strider. Since only genomic data was available for *Proteromonas lacertae,* spliceosomal introns were manually predicted based on homology with *Blastocystis* protein sequences. Sequences were retrieved from the iMicrobe database (Marine Microbial Eukaryotic Transcriptome sSquencing Project)(https://www.imicrobe.us/#/projects/104) and in-house sequencing projects for *Mastigamoeba balamuthi, Blastocystis hominis, Condylostoma magnum* (provided by Dr. Eleni Gentekaki and Dr. Denis Lynn) and *Copromyxa protea* (provided by Dr. Matthew Brown). These sequences were aligned with MAFFT-linsi 7.273 (*Katoh and Toh, 2008*) and used as a seed alignment for building a hidden markov model (HMM) using hmmer3 (*Eddy, 1998*). This HMM model was used to query the *nr* database (Genbank, February 2018) using hmmsearch. To remove false positives, the resulting sequences were used as queries for hmmscan using a modified Pfam database (supplemented with the RquA hmm), and only those sequences that had a reciprocal best hmmscan hit to RquA were kept for further analyses.

Mitochondrial targeting sequences were predicted for each sequence using MitoProt, TargetP v1.1, and MitoFates v1 (*Claros and Vincens, 1996*; *Emanuelsson et al., 2000*; *Fukasawa et al., 2015*). Sequences were tentatively annotated as 'mitochondrial' if two or more software programs predicted mitochondrial localization scores greater than 0.5. The gene context of *rquA* in bacterial genomes was determined by manual investigation of the relevant genome sequences deposited in GenBank (*Supplementary file 1*).

## Phylogenetic analyses

Alphaproteobacterial proteins were aligned based on their HMM profiles in the Phyla-AMPHORA database and ambiguously aligned regions removed using Zorro as implemented in Phyla-AMPHORA (*Wang and Wu, 2013*). Initial single gene trees were constructed using the LG4X model implemented in IQ-TREE 1.5.5 and manually inspected for in-paralogues. These 200 marker genes were concatenated to generate a supermatrix of 54 400 sites and 210 taxa. An initial phylogenetic inference was performed with IQ-TREE v1.5.5 under the LG4X model, followed by reanalysis with the using the model LG + C60 + F (PMSF)+ $\Gamma$4 and using a guide tree inferred with the LG4X model (*Minh et al., 2013*; *Nguyen et al., 2015*; *Wang et al., 20172018*). Datasets are available at on dryad at DOI: https://doi.org/10.5061/dryad.qp745/4.

For the RquA analyses, sequences were aligned using MAFFT-linsi version 7.273 (*Katoh and Toh, 2008*) and regions of ambiguous alignment were removed using BMGE 1.12 (*Criscuolo and Gribaldo, 2010*) with default settings. Evolutionary model selection for maximum-likelihood analysis was performed on all C-series mixture models (C10-C60) with the LG exchangeability matrix, with and without the four-category discrete gamma distribution (+$\Gamma$), invariable sites (+I) and empirical amino acid frequencies (+F) options. The best-scoring models under the Akaike Information Criterion (*Posada and Buckley, 2004*) corrected for small sample size (AIC$_c$) were (i) LG + C60 + $\Gamma$ for the Group A + Group B data set and Group A data set and (ii) LG + C60 + $\Gamma$ + I for the Group B data set. Phylogenies were estimated with IQ-TREE 1.5.5 (*Nguyen et al., 2015*) with branch support estimated by 1000 ultra-fast bootstrap replicates. Bayesian inference was conducted using PhyloBayes 3.2 (*Lartillot et al., 2009*) by running four Markov chain Monte Carlo (MCMC) chains (-catfix C20, -poisson options).

Previous reports hypothesized that RquA evolved from a methyltransferase (UBIE) involved in UQ biosynthesis (*Lonjers et al., 2012*). Indeed, initial phylogenies of RquA and UBIE sequences revealed RquA to emerge from within a larger clade of UBIE sequences (SuppFile 1). Bootstrap support for branches was determined from a total of 1000 ultrafast bootstrap replicates and values were mapped onto the best-scoring ML tree. MCMC chains were run sampling every 10th tree until all four chains converged with a maximum-difference (max-diff = 0.049) with maximum discrepancy less than 0.05 and effective size estimates greater than 3300 for each parameter as calculated with trace-comp. The final consensus tree with posterior probabilities was generated from 2800 trees with a manually determined burn-in of 2000, sampling every 2nd tree. Posterior probabilities (PP) for splits were mapped onto the ML topology estimated with IQ-TREE, using the Dendropy package (*Sukumaran and Holder, 2010*). Datasets are available at on dryad at DOI: https://doi.org/10.5061/dryad.qp745/4.

We tested multiple topologies for the full RquA, Group A and Group B datasets (*Supplementary files 2–4*) using IQ-TREE and CONSEL. Briefly, maximum likelihood trees for various topologies were generated in IQTREE. Using the ML tree, the constrained trees, and 100 ultra-fast bootstrap trees, we computed the site log likelihood values with IQ-TREE (-wsl option) and performed the approximate unbiased test using CONSEL (*Shimodaira and Hasegawa, 2001*). Topologies with an AU p-value less than 0.05 were rejected. Datasets are available at on dryad at DOI: https://doi.org/10.5061/dryad.qp745/4.

## Identification of quinone-utilizing enzymes in eukaryotes

Sequences from respiratory chain complexes (CI, CIII, CIV), quinone biosynthesis enzymes (COQ1-10), alternative oxidase (AOX), dihydroorotate dehydrogenase, sulfite:quinone reductase, electron transferring flavoprotein, (ETFα and β) and ETF dehydrogenase (ETFDH) were manually retrieved from *Arabidopsis thaliana*, *Dictyostelium discodeum*, *Saccharomyces cerevisiae* and *Escherichia coli* the Kyoto Encyclopedia of Genes and Genomes. These Q-utilizing enzyme sequences were used as queries to search each eukaryotic genome or transcriptome than also encoded *rquA* using BLAST or TBLASTN. We tried to identify additional genes that could be linked to *rquA* function using phylogenomic profiling methods. We looked for genes shared with different combinations of *rquA*-containing organisms that are not found in *rquA*-deficient organisms (e.g. *E. coli*, yeast) using the phylogenomic profiling toolkit at the Joint Genome Institute (https://img.jgi.doe.gov/cgi-bin/m/main.cgi?section=PhylogenProfiler&page=phyloProfileForm).

## Genetic linkage in bacteria

When possible, the genomic record for each bacterial genome was retrieved from GenBank via the e-utilities toolkit (https://www.ncbi.nlm.nih.gov/books/NBK25501/) in GTF format using an in-house python script. Accession numbers for the 15 neighbouring genes upstream and downstream of *rquA* were used to retrieve the protein sequences of each of these genes. PFAM domains were assigned to each of these proteins using hmmscan in the hmmer3 (http://hmmer.org) (*Eddy, 1998*). Neighbouring genes were manually examined and genes related to respiration (e.g., electron transport complexes) or associated proteins (e.g., cytochrome, heme and ubiquinone metabolism) were annotated in *Supplementary file 3*).

## Lipid extraction and mass spectrometry

*Pygsuia biforma* cells or bacterial (*Pygsuia*'s bacterial prey or *Rhodospirillum rubrum*) were collected by centrifugation at 500 x g for 10 min, or 14 000 x g for 5 min, respectively at 4°C. Cell pellets were resuspended in 2 mL of methanol before adding 2 mL of petroleum ether (with 10 µM butylated hydroxytoluene, Sigma), vortexed vigorously, and separated by centrifugation at 1000 x g for 5 min. Etherial layers were collected and a second extraction with 2 mL of petroleum ether was performed. The ether extracts were combined and evaporated under $N_2$. The resulting extract was resuspended in 100 µL of ethanol and filtered using a spin column equipped with a 100,000 MWCO filter (Millipore).

Initial method development of the liquid chromatograph (LC) and mass spectrometer (MS) parameters was performed using $UQ_{10}$ (400 pg; Sigma), crude quinone extracts from *R. rubrum* and synthetic $RQ_3$ (400 pg) standards (*Lonjers et al., 2012*). Optimal separation of lipid components was

observed using a 60 min gradient of increasing acetonitrile (ACN) concentrations (60%, 5 min; 60–99%, 30 min.; 99%, 5 min; 99–60%, 2.5 min; 60%, 17.5 min) using a flow rate of 15 µL/min on a Ultimate 3000RS LC nano system (Thermo Scientific, Fremont, CA) using a Jupiter C4 column (Phenomenex, Torrence, California; $150 \times 0.50$ mm, 00F-4167-AF) coupled to a triple quadrupole tandem mass spectrometer (QTRAP 5500,AB Sciex, Concord, Ontario, Canada) equipped with a TurboIonSpray heated electrospray source (Sciex). The source temperature was set at 50°C, declustering potential was set at 80 V, collision energy was set to 20, CAD gas to High and the Gas 1 (nitrogen) nebulizer parameter at 13 (arbitrary units). To determine the diagnostic fragmentation product ions of the RQ or UQ species, we infused RQ3 or UQ10 standards directly into the MS and acquired the product ion (MS/MS) spectra of their precursor ions at $m/z$ 372 and 863, respectively (*Figure 4—figure supplement 1*). From these spectra we were able to detect two predominant diagnostic product ions at $m/z$ 182 and 197, corresponding to the head groups of $RQ_3$ and $UQ_{10}$, respectively. These product ions are diagnostic to the RQ and UQ series, regardless of their chain length. Other product ions were also detected, confirming the structure of the standards, but discarded for our subsequent analyses due to their low intensity. We next developed a Selected Reaction Monitoring method (SRM) taking into account the precursors ions of both UQ and RQ series with chain lengths of 8, 9 and 10 repetitive isoprenylated units as follows. Transitions for $RQ_{10}$, $RQ_9$, and $RQ_8$, were 848.7, 780.6, and 712.6 m/z yielding 182.1 m/z daughter ion corresponding to the RQ head group; transitions for $UQ_{10}$ $UQ_9$ $UQ_8$ 863.7, 795.6, and 727.6 m/z yielding a 197.1 m/z daughter ion corresponding to the UQ head group. We next analyzed the lipid components of wild type *R. rubrum* by LC-SRM-MS/MS to determine the composition of RQ or UQ species in each one of our samples. In general, the different isoprenylated chain length species eluted in 3 min intervals and RQ species eluted ~1 min before the analogous UQ species, consistent with the expected retention times according to the standards and chain lengths. With these elution times and fragmentation patterns as a reference we analyzed the lipid content of *Pygsuia biforma* or its bacterial prey.

## Acknowledgements

This research, and CWS and LE, were supported by a grant from the Canadian Institutes of Health Research (CIHR) MOP 142349 awarded to AJR. CWS was also supported by the Natural Sciences and Engineering Council of Canada Canadian Graduate Studentship (NSERC-CGS-D) and the Killiam Trusts (Level 2). JNS was supported by the National Institutes of Health (award no. 1R15GM096398-01). Any opinions, findings, and conclusions or recommendations expressed in this material are those of the authors and do not necessarily reflect the views of the National Institutes of Health. Work in the GD laboratory was supported by an NSERC Discovery Grant (RGPIN 05616). Work from JPF laboratory was supported by a CIHR (MOP 341174). The authors would also like to acknowledge Dr. Dale Corkery for assistance in microscopy; and Drs. Eleni Gentekaki (Mae Fah Luang University), Denis Lynn (University of Guelph), and Matthew Brown (Mississippi State University) for providing sequences; Dr. Barbara Karten (Dalhousie University) for providing equipment and expertise in lipid extractions; Dr. Melanie Dobson (Dalhousie University) for providing pGEX plasmids; and Dr. Joran Martijn for discussion on alphaproteobacterial phylogeny.

## Additional information

### Funding

| Funder | Grant reference number | Author |
| --- | --- | --- |
| Canadian Institutes of Health Research | MOP 142349 | Andrew J Roger |
| National Institutes of Health | 1R15GM096398-01 | Jennifer N Shepherd |
| Natural Sciences and Engineering Research Council of Canada | NSERC-CGS-D | Courtney W Stairs |
| Killam Trusts | | Courtney W Stairs Courtney W Stairs |

| Natural Sciences and Engineering Research Council of Canada | RGPIN 05616 | Graham Dellaire |
| Canadian Institutes of Health Research | MOP 341174 | James P Fawcett |

The funders had no role in study design, data collection and interpretation, or the decision to submit the work for publication.

### Author contributions

Courtney W Stairs, Conceptualization, Data curation, Formal analysis, Investigation, Visualization, Methodology, Writing—original draft, Project administration, Writing—review and editing; Laura Eme, Conceptualization, Data curation, Formal analysis, Methodology, Writing—original draft, Writing—review and editing; Sergio A Muñoz-Gómez, Formal analysis, Investigation; Alejandro Cohen, Resources, Formal analysis, Investigation, Methodology, Writing—review and editing; Graham Dellaire, Resources, Visualization, Writing—review and editing; Jennifer N Shepherd, Resources, Methodology, Writing—review and editing; James P Fawcett, Resources, Supervision, Project administration, Writing—review and editing; Andrew J Roger, Conceptualization, Resources, Supervision, Funding acquisition, Methodology, Writing—original draft, Project administration, Writing—review and editing

### Author ORCIDs

Courtney W Stairs http://orcid.org/0000-0001-6650-0970
Graham Dellaire http://orcid.org/0000-0002-3466-6316
Andrew J Roger http://orcid.org/0000-0003-1370-9820

### Decision letter and Author response

Decision letter https://doi.org/10.7554/eLife.34292.027
Author response https://doi.org/10.7554/eLife.34292.028

## Additional files

### Supplementary files

• Supplementary file 1. Excel file with workbooks containing information about the number of genomes surveyed, PFAM domains of putative RquA homologues, gene accession numbers for RquA and Q-utilizing proteins, mitochondrial targeting sequence information, redox half-potentials used for *Figure 5*, bacterial genes with and without linkage to respiratory complexes.
DOI: https://doi.org/10.7554/eLife.34292.017

• Supplementary file 2. PDF with all the trees for the full RquA phylogenetic analysis and associated topology tests
DOI: https://doi.org/10.7554/eLife.34292.018

• Supplementary file 3. PDF with all the tree for the Group A phylogenetic analysis and associated topology tests
DOI: https://doi.org/10.7554/eLife.34292.019

• Supplementary file 4. PDF with all the tree for the Group B phylogenetic analysis and associated topology tests
DOI: https://doi.org/10.7554/eLife.34292.020

• Transparent reporting form
DOI: https://doi.org/10.7554/eLife.34292.021

### Data availability

All data is available on Dryad DOI: https://doi.org/10.5061/dryad.qp745

The following dataset was generated:

| | Database, license, and accessibility |

| Author(s) | Year | Dataset title | Dataset URL | information |
|---|---|---|---|---|
| Stairs CW, Eme L, Sergio A Muñoz-Gómez, Cohen A, Dellaire G, Shepherd JN, Fawcett JP, Roger AJ | 2018 | Data from Microbial eukaryotes have adapted to hypoxia by horizontal acquisitions of a gene involved in rhodoquinone biosynthesis | http://dx.doi.org/10.5061/dryad.qp745 | Available at Dryad Digital Repository under a CC0 Public Domain Dedication |

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
