## [Decision Letter]

Thank you for submitting your article "Microbial eukaryotes have adapted to hypoxia by a horizontal acquisition of a gene involved in rhodoquinone biosynthesis" for consideration by *eLife*. Your article has been favorably evaluated by Gisela Storz (Senior Editor) and three reviewers, one of whom is a member of our Board of Reviewing Editors. The following individual involved in review of your submission has agreed to reveal her identity: Julie Dunning-Hotopp (Reviewer #2).

The reviewers have discussed the reviews with one another and the Reviewing Editor has drafted this decision to help you prepare a revised submission.

Summary:

All reviewers think this is an interesting paper, and all were particularly impressed with the author's microscopy and biochemical work showing that the gene in question was localized to the organelle and that the metabolite profile was consistent with the predicted function. Overall the reviewers thought this paper will make a valuable contribution to the literature. It is a clear example of an HGT having a critically important role in the evolution of the anaerobic lifestyle, and it adds to a fairly short list of inbound eukaryotic HGTs that have been shown to have a clear and specific function. Five major points were raised, however, that need to be addressed in a revision.

Essential revisions:

1) This paper argues that *rquA* has been laterally transferred between bacteria and eukaryotes in some combination. The precise origin of the enzyme and the direction of transfer are unclear at present. While some reviewers thought the case for HGT was convincing and carefully and conservatively argued, there was some concern about the interpretation of the patchy appearance of *rquA* in the eukaryotic tree. In particular, there does not appear to be a large amount of sequence data (genome or transcriptome) from these organisms in the databases. Please try to quantify, or at least explain, when there has been enough sequencing to expect the gene to be recovered (for example, see https://www.biorxiv.org/content/early/2017/12/05/229468).

2) Some reviewers wanted more discussion about the possibility of gene loss generating the patterns seen in *rquA* distribution. In particular, it seems plausible that this gene was present in the ancestor of all eukaryotes, and has been lost in aerobic eukaryotes. Could the taxonomic distribution be explained by the gene being present in the ancestor of all eukaryotes but one significant, key loss in a lineage like Opisthokonts? The current taxonomic disparities could then be explained as eukaryote to bacteria LGT, as well as other explanations. The idea of eukaryote-to-bacteria LGT may have been addressed in the second paragraph of the subsection “Phylogenetic Analysis of Bacterial and Eukaryotic RquA Homologs”, however there is some concern that the prokaryotic genes may be confounding the tests.

3) Please address whether some of these *rquA* genes might have been obtained from the mitochondrial endosymbiont. There are a number of alphaproteobacterial sequences in the tree, and it would be useful to test whether an alphaproteobacterial origin of the eukaryotic sequences can be rejected (e.g. taking the closest alphaproteobacterial sequence or clade in each case).

4) All three reviewers were concerned that the genome/transcriptome and database searches performed with *rquA* were too insensitive. BLASTP is a great start but what should be done is to build an HMM with diverse examples of the protein to search more sensitively (using HMMER, most likely), and perhaps iterate and rebuild the HMM if new examples are found. The reviewers don't think this will change the fundamental story but *rquA* distribution is an important part of this paper (and it's not much work) so it should be done to give the reader confidence that this gene is indeed sparsely distributed.

5) Some reviewers wanted more detail about the conservation of introns in the eukaryotic versions of *rquA*. Are these introns found at conserved positions within the *rquA* genes (suggesting a single origin and "eukaryotification" of the gene) or at different sites, which would be consistent with multiple independent acquisitions by LGT?

---

## [Author Response]

Essential revisions:1) This paper argues that rquA has been laterally transferred between bacteria and eukaryotes in some combination. The precise origin of the enzyme and the direction of transfer are unclear at present. While some reviewers thought the case for HGT was convincing and carefully and conservatively argued, there was some concern about the interpretation of the patchy appearance of rquA in the eukaryotic tree. In particular, there does not appear to be a large amount of sequence data (genome or transcriptome) from these organisms in the databases. Please try to quantify, or at least explain, when there has been enough sequencing to expect the gene to be recovered (for example, see https://www.biorxiv.org/content/early/2017/12/05/229468).

We thank the reviewers for raising this concern. Indeed, it is difficult to clearly demonstrate the absence of a gene from a transcriptome or a draft genome. To help illustrate the breadth of our search, we have summarized the number of complete genomes and transcriptomes surveyed from the highest level of taxonomy on NCBI (Author response table 1 below and Supplementary file 1).

Unfortunately, we are unable to assess the completeness of the transcriptomes available on the NCBI transcriptome shotgun assembly database (TSA) without performing the completeness estimates (using BUSCO for instance). This would involve retrieving and analyzing over 2000 transcriptomes, and we believe that this is beyond the scope of our study. However, we are able to provide information on the completeness estimates of the MMETSP data from https://github.com/ljcohen/MMETSP. Over 50% of the transcriptomes available on MMETSP are greater than 66% complete according to BUSCO measurements (see Author response image 1).

We have added the following sentences to the Results section to indicate the databases surveyed.

“Within eukaryotes, we identified *rquA* homologs in 21 representatives of four of the five super-groups of eukaryotes (Obazoa, Amoebozoa, Sar, and Excavata;Figure 2—figure supplement 2). […] Note that the absence of *rquA* in some of these data (particularly the transcriptomes) may be due to a lack of depth of sequence sampling.”

**Author response image 1. respfig1:** Cumulative frequency plot of transcriptome completeness for the marine microbial eukaryotes sequencing projects.

**Author response table 1. resptable1:** Numbers of complete genomes and transcriptome projects surveyed from each of the major lineage of eukaryotes.

	RquA detected^a^	NCBI Genomes^b^	NCIB TSA nr^c^	MMETSP^d^
Alveolata	7	105	32	134
Amoebozoa	4	41	10	8
Apusozoa	0	1	0	0
Breviatea	1	1	1	0
Centroheliozoa	0	0	1	0
Cryptophyta	0	8	1	25
Euglenozoa	2^e^	55	3	5
Fornicata	0	2	2	0
Glaucocystophyceae	0	4	0	3
Haptophyceae	0	5	2	61
Heterolobosea	0	5	1	2
Jakobida	0	6	0	0
Malawimonadidae	0	2	0	0
Opisthokonta	1	9,240	1395	10
Oxymonadida	0	1	1	0
Parabasalia	0	2	1	0
Rhizaria	4	12	4	21
Rhodophyta	0	124	15	8
Stramenopiles	5	150	49	279
Viridiplantae	0	2,461	499	72
unclassified	0	0	3	19
Multispecies with undefined taxa ID	0	0	82	19
Total	24	12225	2102	647
a Source: this study b Source: NCBI Taxonomy Browser, selecting “Genomes” – January 28, 2018 c Source NCBI Trace Archive for the transcriptome shotgun assembly database availble at https://www.ncbi.nlm.nih.gov/Traces/wgs/?term=tsa Taxonomy ID were extracted and taxonomy parsed with ete-toolkit. Only non-redundant taxonomy IDs are shown.d Source: iMicrobe; Taxonomy ID were extracted and taxonomy assigned with ete-toolkit. Only non-redundant taxonomy IDs are shown. e Three copies were identified in Eutriptiella and 1 in Euglena

2) Some reviewers wanted more discussion about the possibility of gene loss generating the patterns seen in rquA distribution. In particular, it seems plausible that this gene was present in the ancestor of all eukaryotes, and has been lost in aerobic eukaryotes. Could the taxonomic distribution be explained by the gene being present in the ancestor of all eukaryotes but one significant, key loss in a lineage like Opisthokonts? The current taxonomic disparities could then be explained as eukaryote to bacteria LGT, as well as other explanations. The idea of eukaryote-to-bacteria LGT may have been addressed in the second paragraph of the subsection “Phylogenetic Analysis of Bacterial and Eukaryotic RquA Homologs”, however there is some concern that the prokaryotic genes may be confounding the tests.

We understand the reviewers concerns. We note that any potential case of LGT can *always* be explained by an alternative vertical descent scenario that involves ancestral presence, gene duplications and massive differential loss. So we suggest that the question is not whether RquA in eukaryotes *can* be explained by such a scenario, but rather what is the most likely explanation for the origins of these genes in eukaryotes. Our analyses show unequivocally that there are two robust clades of RquA: A and B. There are eukaryotic sequences and multiple bacterial sequences that branch within each of these clades. The chances that RquA is ancestral to all eukaryotes seems quite remote given the extreme rarity and patchiness of the gene; of 12225 eukaryotic genomes (in NCBI), 2102 genome shotgun projects (in NCBI TSA database) and 647 transcriptomes in the MMETSP only 24 eukaryote homologs of the RquA gene were found. The eukaryotes that encode the RquA protein are lineages adapted to permanent or transient anoxia that emerge from within predominantly aerobic groups in the global eukaryotic tree. Furthermore, they are distantly related to one another. It is therefore very unparsimonious to posit of all of the ancestors of these organisms going back to the last eukaryotic common ancestor (LECA) retained this gene but, then after these lineages diverged from other eukaryotes (some quite recently), there was widespread loss in dozens of disparate eukaryotic lineages giving rise to the gene presence/absence patterns observed. For example, even if transcriptome data were discounted, for group A RquA enzymes, secondary loss events would have had to happen in *many*different eukaryote lineages for which complete genome data is available. Assuming for the sake of argument a ‘Opimoda/Diphoda’ root for the tree of eukaryotes (Derelle et al. (2015) *Proc. Natl. Acad. Sci.* USA 112:E693-699), the secondary loss list would include at least several Amoebozoan lineages except *Mastigamoeba* and neoparamoebids (e.g. *Acanthamoeba, Entamoeba*, dictyostelids, *Physarum*), other lineages of stramenopiles besides *Blastocystis/Proteromona*s (e.g. diatoms, *Ectocarpus, Phytophthora*), the common ancestor of all opisthokonts, *Thecamonas*, other lineages of Rhizaria besides *Brevimastigamonas* and foraminifera (e.g. *Bigelowiella*) and common ancestors of Heteroloboseans (e.g. *Naegleria*), Metamonada (*Giardia, Trichomonas, Monocercomonoides*), Alveolates, Cryptophytes (e.g. *Guillardia*) and Archaeplastida. The corresponding list for group B *rquA* is similarly long. Both lists get considerably longer if the absence of the gene from transcriptome data for diverse eukaryotes is considered as evidence.

Furthermore, a strictly vertical descent and gene duplication scenario would require that the ancestral eukaryote (LECA) and the mitochondrial endosymbiont (assuming a mitochondrial origin) had at least two copies of the *rquA* gene (at least one of the A family and one of the B family). No organism (prokaryote or eukaryote) currently known (out of 182 organisms *rquA* (166 organism with *rquA* sequences less than 90% identical) encodes both an A-type and B-type in their genome, so this scenario also seems rather unlikely. The idea that eukaryotes donated some of the genes to bacteria also seems implausible because most of the *rquA*-containing eukaryotes for which genomic data is available possess spliceosomal introns in their *rquA* genes. Such introns provide a barrier to successful eukaryote-to-prokaryote gene transfer (at least if transfer occurs on the DNA level).

For these reasons it is more plausible, in our view, that recent independent gains of the gene occurred by prokaryote-to-eukaryote LGT followed by eukaryote-to-eukaryote LGT events during adaptation of these organisms to facultative anaerobiosis.

We have substantially revised the Discussion section of our manuscript to try to clarify our reasoning and include the foregoing arguments.

3) Please address whether some of these rquA genes might have been obtained from the mitochondrial endosymbiont. There are a number of alphaproteobacterial sequences in the tree, and it would be useful to test whether an alphaproteobacterial origin of the eukaryotic sequences can be rejected (e.g. taking the closest alphaproteobacterial sequence or clade in each case).

We investigated whether some of the *rquA* genes might have been obtained from the mitochondria symbiont by doing two kinds of analyses:

Analyses (1): We performed a number of additional topology tests. For the full dataset we tested: (i) the monophyly of eukaryotes + all alphaproteobacteria, (ii) the monophyly of Group A eukaryotes + Group A alphaproteobacteria, and the monophyly of Group B eukaryotes + Group B alphaproteobacteria. Each of these topologies was rejected with a p-value << 0.01.

Our further mining of databases (point #4 below) yielded some metagenome-assembled genomes (MAGs) of new alphaproteobacteria (*Candidatus* ‘Azospirillum’ and *Candidatus* ‘Acetobacter’ metagenome clusters that we refer to as ‘MAG Azo/Aceto alphaproteobacteria’). In our new phylogenies this new group branched closely to a number of Group A eukaryote sequences (e.g. *Blastocystis, Proteromonas,* neoparamoebids, euglenids and *Pygsuia* that we refer to as ‘Group A1 eukaryotes’). In any case, for the full data set we tested the specific affinity of all Group A eukaryotes (A1 and others in group A) to MAG alphaproteobacteria specifically and could not reject this topology (p = 0.227).

We further tested the affinities of the various taxa using Group A- and Group B-specific data sets. As expected, the Group A eukaryotes + MAG Azo/Aceto alphaproteobacteria topology was not rejected, but the Group A eukaryotes + all Group A alphaproteobacteria (including MAGs) topology was strongly rejected (p-value = 3 x 10^-59^). For Group B there were no alphaproteobacterial lineages particularly close to the eukaryotes in the phylogeny (eukaryotes tend to group with disparate betaproteobacteria in different parts of Group B), and the test of the Group B eukaryotes + all Group B alphaproteobacteria topology resulted in strong rejection (p-value = 2 x 10^-75^).

Conclusions from (1): Topologies that have *all* the alphaproteobacteria grouping together in a clade with eukaryotes from both Groups A and B, or individually within Group A or Group B are firmly rejected by topology tests. Note that such monophyletic eukaryotes + alphaproteobacterial topologies (or topologies consistent with them) are usually recovered for alphaproteobacterial-derived mitochondrial proteins (e.g. chaperones, PDH subunits, ETC subunits, FtsZ *etc.*). Group A and Group B eukaryote homologs (and alphaproteobacterial homologs in these groups) clearly have distinct origins from each other. However, the possibility that eukaryotes are monophyletic within each of these groups cannot be rejected. Group A eukaryote homologs could have originated from transfer from an ancestral ‘MAG Azo/Aceto’ alphaproteobacteria specifically. This would be evidence of a mitochondrial origin only if these alphaproteobacteria (a sub-group of Rhodospirillales) were candidate for mitochondrial sister groups. However, phylogenomic analyses do not implicate these taxa as mitochondrial sisters; instead they suggest mitochondria are related to either Rickettsiales, Holosporales, Pelagibacteriales or as a deep independent branch of alphaproteobacteria. It seems, therefore, unlikely that the Group A eukaryotes + MAG Azo/Aceto alphaproteobacteria relationship is suggestive of a mitochondrial origin. Group B eukaryotic homologs have no specific affinities to alphaproteobacteria and their alphaproteobacterial origin seems unlikely.

Analyses (2): If either Group A or Group B eukaryotic *rquA*s originated from the mitochondrial endosymbiont, then the only positive evidence for this would be if these genes were either widespread and ancient within the alphaproteobacterial tree and/or the genes was present in the genomes of the closest alphaproteobacterial relatives to mitochondria. To address this we assembled a phylogenomic matrix of 200 conserved ‘core’ proteins from this phylum to place the *rquA*-containing alphaproteobacterial taxa from our analyses within the context of a representative alphaproteobacterial species tree (note that the tree was constructed in such a way to always represent *rquA*-containing taxa, but the remainder of taxa were sub-seletcted from available genomes to maximize diversity). That phylogeny (now shown as Figure 2—figure supplement 2) shows that many *rquA*-containing alphaproteobacterial genomes occur patchily as derived lineages throughout the various order of the alphaproteobacterial tree. Group A- and Group B-containing taxa are interspersed. The extreme rarity of the enzyme within each of the major α groups for which genomes are available is also shown in this figure. The frequency of *rquA*-containing genomes in each alphaproteobacterial order ranged from 0% (for 5 out of 8 orders) to 9.6% (Rhodospirillales).

Conclusions from (2): Based on the extreme rarity and patchy distribution of *rquA* across alphaproteobacteria it is very unlikely that either the A-type or B-type gene was encoded in ancestral alphaproteobacterial genomes. Furthermore, no alphaproteobacteria (nor *any* organism so far known) encodes both an A-type or a B-type enzyme, making a single mitochondrial origin for both types very unlikely. The candidate sister groups to mitochondria based on phylogenomic evidence are the Rickettsiaciae, Holosporales, Pelagibacteriales or mitochondria emerge as the deepest branch in the alphaproteobacterial tree (see Box 1 of Roger et al. (2017) Curr. Biol.27: R1177-R1192 and references therein). None of these first three groups contain *rquA*-encoding species, and, since it is unlikely that *rquA* is an ancestral alphaproteobacterial gene, there is no positive evidence for a mitochondrial origin of *rquA* in eukaryotes.

As a result of these analyses, we have substantially changed the sections of the Results discussing the topology tests (Phylogenetic Analysis of Bacterial and Eukaryotic RquA Homologs)and added a new section discussing Analysis (2) above entitled: “The distribution of *rquA* amongst alphaproteobacteria”.

We also now explicitly discuss the implications of analyses (1) and (2) in the Discussion.

4) All three reviewers were concerned that the genome/transcriptome and database searches performed with rquA were too insensitive. BLASTP is a great start but what should be done is to build an HMM with diverse examples of the protein to search more sensitively (using HMMER, most likely), and perhaps iterate and rebuild the HMM if new examples are found. The reviewers don't think this will change the fundamental story but rquA distribution is an important part of this paper (and it's not much work) so it should be done to give the reader confidence that this gene is indeed sparsely distributed.

We thank the reviewers for this helpful suggestion. To address this concern, we used our existing RquA alignment and masked N-terminal and C-terminal regions to generate a specific RquA hmmer profile (RquA.hmm). We then generated profiles for COG2226 (the best scoring domain for many RquA and related UBIE/methyltransferase homologues on NCBI conserved domains [NCBI CDD]). To capture sequence diversity we generated two COG2226 profiles: (i) using the NCBI CDD alignment and (ii) retrieved the UniRef90 records for COG2226 and generated an alignment resulting in two HMMs: COG2226_CDD.hmm and COG2226_UniRef90.hmm. Using the RquA.hmm profile and hmmsearch, we retrieved 22174 protein sequences from NCBI GenBank (February 7 2018). We then performed a hmmscan of these 22174 against a custom database of PFAM, RquA and COG2226 profiles. Of these 22174 proteins, only 284 sequences had a best scoring hit to the RquA-specific profile (summarized in Supplementary file 1), 245 of which were deposited at the time of our original analysis (September 2017). Of the 284, the majority represent sequences that are >90% identical to sequences in our existing dataset. However, we did identify some new sequences that were not within this similarity threshold (n=79). We generated a phylogenetic tree of these new sequences together with UBIE and RquA sequences and found that many of the HMM-retrieved sequences belonged to non-RquA clades (Supplementary file 2 – Tree 1; *Mycobacterium* sequences were excluded since they were extremely divergent and could lead to long-branch related artefacts).

Throughout the procedure, we noticed that some of the HMM-retrieved sequences did indeed branch within the RquA clade and so we reanalysed the phylogenetic history of RquA using these newly identified sequences. In particular, we observed six sequences from metagenomics sequencing projects belonging to genomes annotated as *Candidatus* ‘Azospirillum’ and ‘Acetobacter’ species discussed in the answer to point 3 above.

5) Some reviewers wanted more detail about the conservation of introns in the eukaryotic versions of rquA. Are these introns found at conserved positions within the rquA genes (suggesting a single origin and "eukaryotification" of the gene) or at different sites, which would be consistent with multiple independent acquisitions by LGT?

To address this concern, we have generated an alignment indicating where in the eukaryotic sequences we detected introns in genomic data (Supplementary file 3). We observed that closely related sequences (e.g., 2 *Blastocystis* subtypes and *Proteromonas*) shared conserved intron positions – although *Blastocystis* subtype 1 has lost the third intron. However, we were unable to observe conservation of intron positions in any of the other species. We were unable to find introns in the genomic *rquA*sequence from *Stentor*. However, the *Stentor* genome has a low intron density with only 9325 introns identified in the 34506 gene models Slabodnick et al. (2017) Curr. Biol. 20:569-575.

This result is consistent with multiple interpretations of the origins of these proteins. It is well known that intron loss can occur via reverse transcription of spliced messages and recombination with the parental genomic copy. Intron gain can subsequently occur at different intron positions in the same gene (e.g. see Huff et al. (2016) Nature 538:533-536). Thus, both of these phenomena can happen even if the gene evolves by ‘vertical descent’ within eukaryotes. Therefore the lack of intron conservation across the eukaryotic taxa cannot be taken as strong evidence that these homologs had separate bacterial origins, but it is consistent with it.

We have added the following figure and explanation in the Results section:

“We identified spliceosomal introns in the *rquA* genes in eukaryotic taxa for which genomic records were available (i.e., *Proteromonas lacertae, Mastigamoeba balamuthi, Brevimastigomonas motovehiculus, Reticulomyxa filosa*, and all the *Blastocystis* subtype genomes) indicating that these are in fact eukaryotic sequences and not prokaryotic contaminants (Figure 2—figure supplement 1). The *Proteromonas* and *Blastocystis rquA* gene sequences showed conservation of intron position and size. But none of the other eukaryotic *rquA*gene sequences (for which genomic sequence was available) shared intron positions.”